# Linking northeastern North Pacific oxygen changes to upstream surface outcrop variations

Sabine Mecking and Kyla Drushka

University of Washington, Applied Physics Laboratory, Seattle, WA, USA

*Correspondence to*: Sabine Mecking (mecking@uw.edu)

**Abstract.** Understanding the response of the ocean to global warming, including the renewal of ocean waters from the surface (ventilation), is important for future climate predictions. Oxygen distributions in the ocean thermocline have proven an
effective way to infer changes in ventilation because physical processes (ventilation and circulation) that supply oxygen are thought to be primarily responsible for changes in interior oxygen concentrations. Here, the focus is on the North Pacific thermocline, where some of the world ocean's largest oxygen variations have been observed. These variations, described as bi-decadal cycles on top of a small declining trend, are strongest on subsurface isopycnals that outcrop into the mixed layer of the northwestern North Pacific in late winter. In this study, surface density time series are reconstructed in this area using observational data only
and focusing on the time period from 1982, the first full year of the satellite sea surface temperature record, to 2020. It is found that changes in annual maximum outcrop area of the densest isopycnals outcropping in the northwestern North Pacific are correlated with interannual oxygen variability observed at Ocean Station P (OSP) downstream at about a 10-year lag. The hypothesis is that ocean ventilation/uptake of oxygen is greatly reduced when the outcrop areas are small and that this signal travels within the North Pacific Current to OSP, with 10 years being at the higher end of transit times reported in other studies. It is also
found that sea surface salinity (SSS) dominates over sea surface temperature (SST) in driving interannual fluctuations in annual maximum surface density in the northwestern North Pacific, highlighting the role that salinity may play in altering ocean ventilation. In contrast, SSS and SST contribute about equally to the long-term declining surface density trends that are superimposed on the interannual cycles.

## 1. Introduction

Ventilation, the renewal of ocean waters from the sea surface, is expected to decrease due to ocean warming and the resulting increases in stratification (Helm et al., 2011; Capotondi et al., 2012; Heinze et al., 2015). During ventilation, water that is transported from the surface mixed layer into the ocean interior also carries along atmospheric gases, including oxygen ($O_2$) and carbon dioxide ($CO_2$), that have exchanged between the ocean and the atmosphere while the water was at the surface. A reduction in ventilation has thus also been linked to deoxygenation in the ocean interior (Keeling et al., 2010) and reduced uptake of
anthropogenic $CO_2$ (Heinze et al., 2015; Franco et al., 2021) by the world's oceans. This has impacts on ocean ecosystems because a reduction in $O_2$ can shift previously well-oxygenated marine environments to hypoxic conditions (Whitney et al., 2007), affecting marine life. A reduction in oceanic $CO_2$ uptake creates a positive feedback to global warming caused by rising atmospheric $CO_2$ levels since it leaves more $CO_2$ in the atmosphere (Friedlingstein, 2015).

The North Pacific thermocline is one of the first regions for which large variations in oceanic $O_2$ concentrations were observed and linked to ventilation changes (Andreev and Kusakabe, et al., 2001; Emerson et al., 2001; Ono et al., 2001; Watanabe et al., 2001; Emerson et al., 2004). Observations from repeat hydrographic sections have shown that $O_2$ changes between decades can be 20

µmol kg$^{-1}$ or more in both the eastern and western North Pacific (Mecking et al., 2008; Takatani et al., 2012; Sasano et al., 2015). Fifty years of time series data from Ocean Station Papa (OSP; 145°W, 50°N) in the Gulf of Alaska in the northeastern North Pacific (black circles in Fig. 1) have indicated that there are bi-decadal $O_2$ variations that occur on subsurface isopycnals in addition to a smaller declining trend associated with ocean warming (Whitney et al., 2007; Cummins and Ross, 2020). Similar variations (of opposite sign) are found to occur at OSP in dissolved inorganic carbon (DIC) and nutrients (Whitney et al., 2007; Franco et al., 2021) since $O_2$ consumption and nutrient/DIC production are linked to each other through stochiometric ratios during the respiration/remineralization process (Anderson and Sarmiento, 1994). The bi-decadal cycles in the Gulf of Alaska roughly correspond to oscillations in $O_2$ and nutrients observed in the western subarctic North Pacific (Ono et al., 2001) with an order 5–10 year lag depending on the density surface (Whitney et al., 2007), likely reflecting the time that it takes for waters formed in the northwestern North Pacific to travel east with the North Pacific Current.

The mechanisms responsible for the observed variability in thermocline $O_2$ content have been investigated in a variety of ways (Ono et al., 2001; Watanabe et al., 2001; Emerson et al., 2001, 2004; Deutsch et al., 2005; Andreev and Baturina, 2006; Mecking et al., 2006; Whitney et al., 2007, Crawford and Peña, 2016; Sasano et al., 2018; Stramma et al., 2020). One suggestion is that surface density variations, particularly variations in the northwest North Pacific late winter outcrop location and area (including complete cessation of outcropping) of the $\sigma_\theta = 26.6$ kg m$^{-3}$ isopycnal, which exhibits some of the largest $O_2$ variations downstream, play a key factor (Emerson et al., 2004; Mecking et al., 2006, 2008). However, correlating the $O_2$ changes with surface forcing mechanisms, such as temperature and wind stress changes associated with the Pacific Decadal Oscillation (Mantua et al, 1997; Deser et al.,1999) remains inconclusive (Mecking et al., 2008). Alternatively, as a result of the 18.6-year lunar nodal tidal cycle, variations in tidal mixing in the narrow passages between the Kuril Islands (45-50°N) separating the open North Pacific from the Sea of Okhotsk may affect surface and near-surface properties in the northwestern North Pacific (Andreev and Baturina, 2006; Yasuda et al., 2006) and cause the bi-decadal $O_2$ cycles observed downstream at OSP (Whitney et al., 2007).

While there remain open questions regarding the driving forces of the ventilation changes in the North Pacific (Sasano et al., 2018), it seems clear that surface/mixed layer density variations in the northwestern North Pacific are important, in particular in late winter when the densest isopycnals outcrop, mixed layers are the deepest, and permanent subduction/ventilation (i.e. transport of oxygenated waters from the mixed layer into the thermocline that is not reversed through mixed layer deepening in the next winter season) takes place (Huang and Qiu, 1994). Modeling studies (Deutsch et al., 2005, 2006; Kwon et al., 2016) have confirmed what was first suspected from observations (Emerson et al., 2001, 2004): physical processes (gyre ventilation and circulation) are dominant in the North Pacific in producing $O_2$ variations in the ocean interior, with contributions from changes in biological respiration rates being minor.

In this paper, we zoom in on the northwestern North Pacific, reconstructing surface density time series using observational data only. Outcrop area time series calculated from the surface density record are used as a simple proxy for ventilation (Kwon et al., 2016) since they only require temperature and salinity data at the surface (compared to more complex data products for full calculation of subduction rates; see e.g. Huang and Qiu, 2004, and Toyama et al., 2015) and it is clear that subduction ceases if a density class stops outcropping. We focus on the time period from 1982, the beginning of the full annual satellite sea surface temperature record, to the present. We compare this surface density record to the $O_2$ time series at OSP in the northeastern North Pacific and examine connections between ventilation changes in the northwest, including the contributions of salinity versus temperature to surface density changes, and $O_2$ changes downstream. This provides a data-based assessment of the hypothesis that

surface density variations are a main driver of $O_2$ changes in the ocean interior. The data and data methods used are described in Sect. 2, results are discussed in Sect. 3, and summary and conclusions are presented in Sect. 4.

**2. Data and Methods**

**2.1 Datasets**

As the basis for the surface density calculation, we use the EN4 dataset (version EN4.2.1) that provides quality controlled subsurface ocean temperature and salinity profiles as well as objective analysis of those data on monthly 1°x1° grids with 42 vertical levels from 1900 to the present (Good et al., 2013). EN4 is based on measurements from the World Ocean Database

(WOD) as well as from the Global Temperature and Salinity Profile Program (GTSPP, from 1990 onward) and the Argo global data assembly centers (GDACS) from 2000 onward. The data going into EN4 are quality controlled using a series of consistent control steps and then gridded based on an iterative optimal interpolation method, taking into account anomalies (relative to climatology) of prior months' data and providing uncertainty estimates (Good et al., 2013). Here, we utilize the version of EN4 with the Gouretski and Reseghetti (2010) bias correction, though the findings are insensitive to the choice of corrections available

with the EN4 data. We use the upper level of the gridded EN4 temperature and salinity fields that are nominally at 5 m depth and refer to them as sea surface temperature (SST) and salinity (SSS) for brevity.

In addition to SST from EN4, we also use SST data from the NOAA Optimal Interpolation SST (OISST) Version 2 High Resolution product (Reynolds et al., 2007; Banzon et al., 2016). The OISST V2 product is based on a blend of satellite and in situ observations,

and is available on a global 1/4°x1/4° grid from September 1981 onward. We average daily fields to produce monthly SST maps, using data from 1982 to 2020. Comparisons with SST from EN4 enable us to examine effects of spatial resolution on surface density patterns.

Satellite salinity data have only been available since 2009 (European SMOS satellite, 2009–present; U.S. Aquarius satellite, 2011–

2015; U.S. SMAP satellite, 2015–present). An optimally-interpolated multi-satellite salinity product that combines data from all three satellites and includes bias corrections based on in-situ data (Melnichenko et al., 2021) compares well to EN4 with nearly identical salinity averages in the northwestern North Pacific (Fig. A1), but is not used for this study due to its relatively short duration (2011–present). Thus, we instead combine the OISST data with EN4 SSS data (interpolated to 1/4°x1/4°) to obtain surface density fields at higher resolution than EN4 alone, acknowledging the caveat that any small-scale density variability observed is a

result of SST patterns only. We refer to this dataset as EN4-OISST. The TEOS-10 scale is used to compute surface potential density from SST and SSS for both the EN4 and EN4-OISST datasets using the Gibbs Seawater toolbox (McDougall and Barker, 2011).

To examine ocean interior oxygen variations, we use data from OSP (145°W, 50°N), which provides one of the longest data time series in the world's ocean, with weather ship data available since 1956 (Whitney et al., 2007; Cummins and Ross, 2020; Ross et

al., 2020). Currently, OSP is occupied ~3 times per year as part of the Canadian Line-P hydrographic cruises from Vancouver Island to OSP. Cruise data is publicly available from the Fisheries and Oceans Canada database (Freeland, 2007; Whitney et al., 2007), where we downloaded the OSP bottle data from 1956 until 2020. For the calculation of vertical profiles of potential density at OSP, in situ temperature and salinity were converted to conservative temperature and absolute salinity using the TEOS-10 scale (McDougall and Barker, 2011), and the OSP oxygen data were then mapped onto a potential density anomaly-time grid using

standard objective mapping techniques (Bretherton et al., 1976; Roemmich, 1983). To put the OSP time series data into spatial context, we also examine repeat hydrography data (vertical sections of temperature, salinity, and oxygen; Talley et al., 2016) from the northeastern North Pacific. The most recent P16N section, a key U.S. repeat section along 152°W (Emerson et al., 2004; Mecking et al., 2008) that is re-occupied every ten years or so, was occupied in 2015 as part of the Global Ocean Ship-based Hydrographic Investigations Program (GO-SHIP), and the P1 section along 47°N (Watanabe et al., 2001; Emerson et al., 2004) ,

also covering parts of the northeastern North Pacific, was occupied in its full extent as part of the Japanese contribution to GO-SHIP in 2014 (Fig. A2) .

**2.2 Outcrop area as ventilation metric**

Variations in surface potential density are examined, based on the EN4-OISST dataset, using the outcrop area of the densest isopycnals to outcrop in the northwestern North Pacific in March ($\sigma_\theta$ = 26.4–26.7 kg m$^{-3}$) as a metric (Fig. 1). These isopycnals are

chosen because they encompass the isopycnal where the largest North Pacific $O_2$ changes have been observed ($\sigma_\theta$ = 26.6 kg m$^{-3}$; Emerson et al., 2004) and because they mark the bottom of the ventilated thermocline. If they stop outcropping or if the outcrop area is reduced, it can be assumed that ventilation, including transport of oxygen from the surface mixed layer on these density surfaces into the ocean interior, is also stopped or reduced during that time (Mecking et al., 2008). Using late winter data (March) is important because this is when surface waters become the densest and permanent subduction from the mixed layer along

outcropping isopycnals takes place, renewing waters in the deepest part of the ventilated thermocline (Huang and Qiu, 1994). While vertical Ekman pumping and lateral induction estimates are needed to fully quantify the amount of water that is transported across the base of the mixed layer, the spatial extent of the surface outcrop area can be used as a proxy for the amount of ventilation taking place, with a larger outcrop area indicating more ventilation and a smaller outcrop area indicating less ventilation (Mecking et al., 2008; Kwon et al., 2016). Since the OISST product represents temperature at the ocean surface, our outcrop area calculations

based on EN4-OISST (and also EN4) rely on surface density rather than density at the base of the mixed layer where the subduction takes place. We note that surface densities are slightly lower than densities at the base of the mixed layer (by ~0.03 kg m$^{-3}$ for individual profiles; Holte and Talley, 2009), but are expected to represent temporal variations in outcrop area accurately. For the purpose of this paper, the terms surface and mixed layer density are used interchangeably.

**3. Results**

**3.1 Temporal variations in outcrop area of densest outcropping isopycnals**

The surface outcrop areas of the $\sigma_\theta$ = 26.4–26.7 kg m$^{-3}$ isopycnals in March show large interannual differences (Fig. 1, red symbols). The $\sigma_\theta$ = 26.7 kg m$^{-3}$ isopycnal (Fig. 1, dark gray bold lines) reaches the surface in the years of maximum outcrop areas (e.g. 1983, 2001; Fig. 1a, c) as well as the years of minimum outcrop areas (e.g. 1996, 2015; Fig. 1b, d), though barely. However, the strong fluctuations in March outcrop area of $\sigma_\theta$ = 26.4-26.7 kg m$^{-3}$ are caused mostly by changes in the outcrop location of the lighter

isopycnals ($\sigma_\theta$ = 26.4–26.6 kg m$^{-3}$), as can been seen for example by large variations in the $\sigma_\theta$ = 26.4 kg m$^{-3}$ outcrop (Fig. 1, light gray bold lines) that can extend significantly to the east of the dateline in high outcrop area years. In 1983 (Fig. 1a) and 2001 (Fig. 1c), the $\sigma_\theta$ = 26.4–26.7 kg m$^{-3}$ outcrop area forms a large, continuous patch extending from east of Japan to east of the dateline and between ~40°N and the Bering Sea (area 2.47x10$^6$ km$^2$ and 3.58x10$^6$ km$^2$, respectively). In contrast, in 1996 (Fig. 1b) and 2015 (Fig. 1d) this area is contracted to small disjointed outcrop patches along 40°N (area 0.592x10$^6$ km$^2$ and 0.305x10$^6$ km$^2$,

respectively), allowing for very little ventilation. A time series of the monthly $\sigma_\theta$ = 26.4–26.7 kg m$^{-3}$ outcrop area in the North Pacific for all years from 1982 to 2020 (Fig. 2, blue line) illustrates that (1) the outcrop area is the indeed largest in March of each year since the isopycnals in this density range all reach the surface then, and that (2) there are distinct March minima in 1995–1997, 2011, and 2014–2015 (outcrop area < 10$^6$ km$^2$), compared to the March outcrop area in the years before/after and a distinct March maximum in particular in 2001 (outcrop area > 3.5x10$^6$ km$^2$). Uncertainties in the March outcrop areas that are calculated based on the salinity and temperature uncertainties reported with the EN4 dataset (Good et al, 2013) using a Monte Carlo approach are 1–2 magnitudes smaller than the outcrop areas (not shown), indicating that the large interannual changes in outcrop area are significant.

To assess how spatial resolution of data and small-scale variability affects the results, we compare EN4-OISST with the lower-resolution EN4 dataset (1°x1°) that is based on EN4 temperature and salinity (Sect. 2.1). The time series of the $\sigma_\theta$ = 26.4–26.7 kg m$^{-3}$ outcrop area from 1982 to 2020 using EN4 (Fig. 2, red line) exhibits larger March peak values than the EN4-OISST dataset (Fig. 2, blue line) in almost every year, likely because SST fields in EN4 are more smoothed and less patchy than the OISST satellite product. But the EN4-derived outcrop areas show the same pattern of maxima and minima as EN4-OISST, indicating that in particular the extreme minima in 1995–1997, 2011, and 2014–2015 and the extreme maximum in 2001 are a robust result. The EN4-OISST and EN4 datasets (Fig. 2), despite the 2001 maximum, also both show a declining trend in the $\sigma_\theta$ = 26.4–26.7 kg m$^{-3}$ outcrop area from 1982–2020 that is discussed further in the next section. We continue to use the EN4-OISST dataset for most of the remainder of the analysis given its higher resolution, knowing that the trends and overall patterns in outcrop area are consistent among the higher and lower resolution datasets.

In order to examine a longer time frame, we also calculate the $\sigma_\theta$ = 26.4–26.7 kg m$^{-3}$ outcrop area for the full EN4 record since 1900 (Fig. A3) in addition to the 1982–2020 data shown in Fig. 2 (red line). Decadal variability in the outcrop area is also present prior to 1982 (Fig. A3), but the nearly constant outcrop area in the earlier part of the record (1900–1925) stands out as somewhat suspect. Since uncertainties reported with the EN4 data do not become larger going back in time to 1900, as one would expect for earlier data, measurement error/bias is likely not fully accounted for. Thus, in the following, we rely on the EN4-OISST dataset (as mentioned above), except when lag correlations are also calculated using the longer EN4 record (Sect. 3.2).

### 3.2 Correlations between outcrop area and O$_2$ time series

Since OSP is downstream of the $\sigma_\theta$ = 26.4–26.7 kg m$^{-3}$ outcrop area in the northwest Pacific (as indicated by streamlines on $\sigma_\theta$ = 26.6 kg m$^{-3}$ in Fig. A2), O$_2$ distributions observed at OSP, mapped onto a potential density anomaly-time grid (Sect. 2.1), are expected to be affected by O$_2$ uptake variability (i.e., ventilation variability) in the isopycnal outcrop region. For the OSP data, we confirm that there are large decadal variations in O$_2$ concentration (Whitney et al., 2007; Cummins and Ross, 2020) on isopycnals near the bottom of the ventilated thermocline (Fig. 3a) where maximum O$_2$ variability, based on repeat hydrography data, has been observed previously on the $\sigma_\theta$ = 26.6 kg m$^{-3}$ isopycnal (Mecking et al., 2008). Fig. 3b illustrates that densities around $\sigma_\theta$ = 26.6 kg m$^{-3}$ are also the isopycnals where the largest decadal variations (and long-term O$_2$ trends) occur at OSP, as indicated by overall maxima in standard deviation and negative trend of O$_2$ over the time period of the OSP record (1956-2020) around $\sigma_\theta$ = 26.5–26.7 kg m$^{-3}$. Maxima in declining O$_2$ trends at OSP have also been reported in this density range by Whitney et al. (2007; $\sigma_\theta$ = 26.5 kg m$^{-3}$) and Crawford and Peña (2016; $\sigma_\theta$ = 26.7 kg m$^{-3}$), each using slightly different methodologies, but also slightly deeper on $\sigma_\theta$ = 26.8 kg m$^{-3}$ (Cummins and Ross, 2020) which does not outcrop in the North Pacific, solidifying that this is the density range of

largest $O_2$ decline or close to it in the northeastern North Pacific. Combining the $O_2$ data on $\sigma_\theta = 26.6$ kg m$^{-3}$ with the time series of annual maximum $\sigma_\theta = 26.4$–$26.7$ kg m$^{-3}$ outcrop area from the EN4-OISST dataset (maxima in blue line each year in Fig. 2), which we use as a proxy for ventilation of the isopycnals at the bottom of the ventilated thermocline, shows that a strong minimum in $O_2$ in 2005–2007 lags the minimum in annual maximum outcrop area in 1995–1997 by about a decade (Fig. 4, solid blue/cyan versus red/magenta lines). Such a lag is expected, assuming it takes about 10 years for surface waters from the outcrop area in the northwestern North Pacific to reach OSP on this isopycnal (Whitney et al., 2007). This transit time is somewhat larger than the 7–8 year west-to-east transit time estimated by Ueno and Yasuda (2003) on $\sigma_\theta = 26.7$ kg m$^{-3}$ from a North Pacific inverse model, but given the differences in data and methods used and the notion that the $O_2$ cycles in the northeastern North Pacific are affected by more than one process (Sasano et al., 2015, 2018), we consider our transit time estimate to be roughly consistent with Ueno and Yasuda's.

Estimation of lagged correlations between $O_2$ on $\sigma_\theta = 26.6$ kg m$^{-3}$ and the annual maximum $\sigma_\theta = 26.4$–$26.7$ kg m$^{-3}$ outcrop area from EN4-OISST, using the unfiltered time series (Fig. 4, solid blue/red lines) as well as filtered data (Fig. 4, solid cyan/magenta lines), also results in the best correlation at a 10-year lag for the $O_2$ record lagging annual maximum outcrop area (r = 0.50 for filtered, detrended data; Fig. 5). Note that the correlation is even better for $O_2$ leading the annual maximum outcrop area (r = 0.59 at a –10-year lag for $O_2$; Fig. 5), which is unphysical though since $O_2$ in the interior ocean does not have an obvious effect on surface density. However, since the $O_2$ variations and possibly the annual maximum outcrop area are associated with bi-decadal cycles, shifting $O_2$ either forward or backward in time by 10 years should result in a strong correlation with the outcrop area. This also explains why there is a maximum negative correlation at a close to zero lag (r = –0.65, Fig. 5).

We note that on isopycnals deeper than $\sigma_\theta = 26.6$ kg m$^{-3}$, the lag between $O_2$ and the annual maximum $\sigma_\theta = 26.4$–$26.7$ kg m$^{-3}$ outcrop area tends to increase, as shown for $\sigma_\theta = 27.0$ kg m$^{-3}$, where the best correlation for $O_2$ lagging outcrop area is at 14 years (Fig. A4) compared to 10 years for $\sigma_\theta = 26.6$ kg m$^{-3}$ (Fig. 5). This finding is consistent with an increase in west-to-east travel times with density (and depth), as reported by Ueno and Yasuda (2003) using similar densities in their model calculations (see their Fig. 7). However, for isoycnals shallower than $\sigma_\theta = 26.6$ kg m$^{-3}$, we do not find a depth trend in the lag between $O_2$ and outcrop area (not shown) because the $O_2$ cycles at OSP on isopycnals between $\sigma_\theta = 26.2$ kg m$^{-3}$ and $\sigma_\theta = 26.6$ kg m$^{-3}$ are in phase (Fig. 3a). We hypothesize that the $O_2$ signal that peaks near $\sigma_\theta = 26.6$ kg m$^{-3}$ (Fig. 3b) is distributed to the lighter (i.e., shallower) isopycnals through vertical mixing as the water travels west to east and the $O_2$ signals at $\sigma_\theta = 26.6$ kg m$^{-3}$ and above are thus in sync with each other. In contrast, for the isopycnals below $\sigma_\theta = 26.6$ kg m$^{-3}$, which correspond to the upper North Pacific Intermediate Water range ($\sigma_\theta = 26.64$–$27.0$ kg m$^{-3}$; Talley et al., 1997), the $O_2$ signals may be dominated by the vertical mixing processes in the northwestern North Pacific that are part of the formation process of the intermediate water (Sasano et al., 2018). The intermediate water travels eastward then at a slower rate than water on $\sigma_\theta = 26.6$ kg m$^{-3}$ (Ueno and Yasuda, 2003), causing the $O_2$ signals to be out of phase with the shallower layers.

In order to examine a longer time frame, we also calculate lagged correlations between $O_2$ on $\sigma_\theta = 26.6$ kg m$^{-3}$ at OSP and the $\sigma_\theta = 26.4$–$26.7$ kg m$^{-3}$ outcrop area based on the entire EN4 record (Fig. A3) using data since 1941 (Fig. A5; since the $O_2$ record at OSP starts in 1956, 1941 is the first data year used for outcrop area when calculating correlations with a maximum lag of 15 years) instead of the EN4-OISST dataset (Fig. 5). Maximum correlations using EN4 also occur at close to +/–10-year lags (Fig. A5). The magnitude of the maximum correlations is smaller than for the EN4-OISST dataset (Fig. 5) which may be a results of the lower

accuracy of the earlier data in EN4. Nevertheless, the maximum correlations are statistically significant, and the conclusion remains consistent that $O_2$ on $\sigma_\theta = 26.6$ kg m$^{-3}$ at OSP lags variations in outcrop area by about 10 years.

230 In addition to decadal scale variability, both the OSP $O_2$ time series on $\sigma_\theta = 26.6$ kg m$^{-3}$ and the northwestern North Pacific annual maximum outcrop area (Fig. 4) exhibit declining long-term linear trends amounting to $-0.53$ μmol kg$^{-1}$ yr$^{-1}$ and $-2.45 \times 10^{10}$ m$^2$ yr$^{-1}$, respectively (dashed lines in Fig. 4). Previous studies have shown that there is a long-term decrease in $O_2$ at OSP (Whitney et al., 2007; Stramma et al, 2020) as well as in surface density in the northwestern North Pacific (Ono et al., 2001; Durack and Wijffels, 2010), the latter of which is consistent with the decreasing trend in outcrop area of the densest outcropping isopycnals

235 ($\sigma_\theta = 26.4$–26.7 kg m$^{-3}$) that we have shown here. The trends in $O_2$ and outcrop area likely occur together for the same reasons that the decadal scale variability in $O_2$ at OSP and outcrop area are correlated (with $O_2$ lagging; see above): as the outcrop area of the $\sigma_\theta = 26.4$–26.7 kg m$^{-3}$ isopycnals declines, less ventilation of these isopycnals takes place and less $O_2$ is transported from the mixed layer into the ocean interior, causing the declining trend in $O_2$ on those isopycnals at OSP.

240 One caveat to the interpretation of the OSP $O_2$ data in combination with the northwestern North Pacific outcrop area time series comes from repeat hydrography data in the area of OSP (Fig. A2). Near the longitude and latitude of OSP (145°W, 50°N), respectively, CLIVAR/GO-SHIP repeat hydrography section differences show continued $O_2$ decline from the mid-2000s to mid-2010s along 152°W (P16N: 2015 minus 2006; Fig. A6a) and along 47°N (P1: 2014 minus 2007; Fig. A6b; see also Kouketsu et al., 2020, their Fig. 4b), contrary to the OSP data, which indicate an increase in $O_2$ on the $\sigma_\theta = 26.4$ kg m$^{-3}$ to $\sigma_\theta = 27.0$ kg m$^{-3}$

245 isopycnals between those points in time (Fig. 3a). To resolve this, we combine $O_2$ values from the CLIVAR/GO-SHIP as well as earlier WOCE sections, averaged zonally on isopycnals over the intervals 40–45°N for P16N and 140–165°W for P1 (intervals shown as black lines in Fig. A6), with the OSP time series directly (Fig. A7). This indicates that (1) the section averages for both P16N (Fig. A7a) and P1 (Fig. A7b) may lead the OSP time series by 2–4 years, suggesting that $O_2$ signals arrived at the hydrographic section locations a few years earlier than at OSP and that they follow perhaps a longer, more eastward pathways

250 from 47°N to OSP than the climatological streamlines in Fig. A2 indicate (Kouketsu et al., 2020), and that (2) the repeat hydrography, due to lack of time resolution, does not fully resolve the temporal variability in the $O_2$ record and misses for example the large $O_2$ minimum observed at OSP in 2007 that leads the 2010s increase in $O_2$. While the 2–4 year lead of the section data is larger than one might expect given the proximity of the sections to OSP, we conclude that the repeat hydrography data (within the limits of decadal observations) are still roughly consistent with the OSP data in terms of observed $O_2$ changes.

255 **3.3 Causes**

**3.3.1 SSS and SST variability**

As outlined in the previous section (Sect. 3.2), variability in oxygen at OSP in the eastern North Pacific can be explained in part by upstream fluctuations in isopycnal outcrop areas in the northwestern North Pacific with a lag of about 10 years (Fig. 4, 5). Here, we assess the drivers of the outcrop area fluctuations by examining the effects of SST and SSS in the northwestern North Pacific

260 (averaged over 40-50°N, 150-170°E; see box in Fig. 1a) on surface density (Fig. 6). Contributions of SST and SSS (Fig. 6a) to annual maximum surface density changes in this area (Fig. 6b) are estimated by determining the annual mean cycle of SST and SSS each from the EN4-OISST dataset (also averaged over 40-50°N, 150-170°E) and then calculating surface density with either SST or SSS fixed (at its annual mean cycle) while allowing the other variable to vary (thin solid lines in Fig. 6a).

The SSS data at the time of annual maximum surface density (Fig. 6a, bold red line), usually late winter, show minima around 1996, 2011, and 2015. These match the years when the annual maximum $\sigma_\theta$ = 26.4–26.7 kg m$^{-3}$ outcrop area has minima (Fig. 2) and annual maximum surface densities are the lowest (Fig. 6b, bold black line). Varying SSS and keeping SST fixed at its annual mean cycle results in annual maximum surface densities (Fig. 6b, red dashed line) that follow the annual maximum surface density (bold black line) closely, which indicates that SSS variations are the main driver of interannual variations in annual maximum surface density in the northwestern North Pacific. This includes a rebound to higher surface densities in recent years that is evident in both the total and the SSS-driven density records (Fig. 6b) and that is caused by the increase in SSS since 2017 (Fig. 6a). In contrast, varying SST and keeping SSS fixed at its annual mean cycle causes much smaller interannual variations in the annual maximum surface density (Fig. 6b, blue dashed line) than observed in the full record (bold black line), affirming the lesser role that SST plays in producing the interannual annual maximum surface density variations in the northwestern North Pacific.

Quantitatively, SSS-driven (SST fixed) and SST-driven (SSS fixed) density variability can explain 85% and 21%, respectively, of the variance in total annual maximum surface density in Fig. 6b. The contributions do not add up to 100% exactly due to the non-linearities in the calculation of density (equation of state), but it is clear that SSS variations have a much larger contribution (about four times). The dominance of salinity over temperature in driving density fluctuations is consistent with the relatively large haline contraction coefficient compared to the thermal expansion coefficient at this latitude, which govern the contributions of salinity and temperature, respectively, to density (McDougall and Barker, 2011). In terms of long-term linear density trends (Fig. 6b, thin solid lines), the SSS-based trend (thin red line, –0.0014 kg m$^{-3}$ yr$^{-1}$) and the SST-based trend (thin blue line, –0.0017 kg m$^{-3}$ yr$^{-1}$) contribute about equally to the linear decrease in total annual maximum surface density (thin back line, –0.0032 kg m$^{-3}$ yr$^{-1}$), indicating that long term surface freshening and warming in the northwestern North Pacific are of similar importance in explaining declining surface densities.

Evidence that SSS variations in the northwestern North Pacific are occurring and playing a major role in determining interannual changes in upper ocean density and related ventilation also comes from a study by Uehara et al. (2014), who found upper layer (0–100m) salinity variability in the western subarctic gyre (WSG; their Fig. 4) similar to the SSS variability reported here (Fig. 6a), with our averaging area (see box in Fig. 1a) encompassing the southern half of their WSG definition. The time period examined in Uehara et al. (2014) is 1950 to 2008, overlapping the time period of our surface time series (1982–2020) by 26 years and showing the strongest surface salinity freshening signals in the WSG in the early 1960s and in the late 1990s, with the latter corresponding to the SSS minimum we find around 1996 (Fig. 6a) and the strong reduction in March outcrop area of $\sigma_\theta$ = 26.4–26.7 kg m$^{-3}$ from 1995–1997 (Fig. 2). Uehara et al. (2014) attribute the upper ocean salinity changes in the WSG to changes in atmospheric forcing such as precipitation and wind, and suggest that these changes are propagated downstream by surface currents into the Bering Sea, the East Kamchatka region, the eastern Sea of Okhotsk and finally the shelves in the northwestern Sea of Okhotsk (see pathways in their Fig. 9) where dense shelf waters are formed due to surface cooling. Along this path, mixing variability associated with the 18.6-year nodal tidal cycle, which has been quoted by several authors as a source of upper ocean variability in the northwestern North Pacific especially around the narrow passages in the Kuril Straits that separate the Sea of Okhotsk from the northwestern North Pacific (Yasuda et al. 2006; Whitney et al., 2007; Osafune and Yasuda, 2013; Stramma et al., 2020), may affect upper layer salinities (Uehara et al., 2014). Based on the finding by Uehara et al. that the WSG is at the upstream end of this surface water pathway, however, we conclude that the SSS and outcrop area variability in the northwestern North Pacific reported here (as well as the associated ocean interior O$_2$ changes at OSP in the northeast) are also due to atmospheric forcing changes rather than the

nodal tidal cycle as has been speculated (Whitney et al., 2007). Nevertheless, there could be feedback loops from the dense shelf waters produced in the Sea of Okhotsk at the downstream end of this path back to the surface waters in the open northwestern North Pacific.

### 3.3.2 Climate indices (PDO, NPGO, and NPI)

The Pacific Decadal Oscillation (PDO) is the dominant mode of variability in the North Pacific Ocean, with the PDO index defined as the leading principal component of monthly SST variability north of 20°N (Mantua et al. 1997). Since much of the variability in annual maximum surface density that we associate with fluctuations in the $\sigma_\theta = 26.4–26.7 \, \text{kg m}^{-3}$ outcrop area and $O_2$ downstream appear to be driven by salinity variations, as illustrated in the previous section (Sect. 3.3.1), a correlation between surface density and the PDO is not necessarily expected. Indeed, we find that the correlation between the PDO and the annual maximum surface density (averaged over 40–50°N, 150–170°E) time series (Fig. A8) is only 0.03. There are periods where the two time series seem to be tracking each other (1999–2005), but also several where they clearly show opposite variations (e.g. 1995–1999, 2013–2018), resulting in a correlation close to zero.

Only somewhat higher correlations exist between the annual maximum surface density and the North Pacific Gyre Oscillation (NPGO) index (Di Lorenzo et al., 2008) and the North Pacific Index (NPI; Trenberth and Hurrell, 1994) with r-values of 0.16 (Fig. A9) and -0.14 (Fig. A10), respectively, suggesting that these indices also describe little of the surface density variability that drives downstream $O_2$ variations at OSP. The NPGO index is defined as the second dominant mode of sea surface height (SSH) anomaly variability in the northeastern North Pacific (Di Lorenzo et al., 2008) whereas the NPI represents area-weighted winter time sea level pressure over the northern North Pacific (30°–65°N, 160°E-140°W), which indicates changes in atmospheric circulation. Note that correlations between climate indices and North Pacific thermocline $O_2$ changes have also been analyzed in different ways (Andreev and Kusakabe, 2001; Ono et al., 2001; Mecking et al., 2008; Stramma et al, 2020), with connections between OSP $O_2$ and the PDO (Mecking et al., 2008; Stramma et al., 2020) as well as the NPGO (Stramma et al., 2020) found to be weak or inconclusive, but that we focus here on the correlations of climate indices with northwest North Pacific surface density changes because we are evaluating surface density changes as a driving force for OSP $O_2$ changes in this paper. However, given the dominance of SSS in driving annual maximum surface density changes and the relatively low correlations between the surface density changes and the PDO, NPI or NPGO, we conclude that a climate index that better incorporates salinity, for example through inclusion of SSS and/or E–P (evaporation minus precipitation) data since E–P variability is the main driver for SSS changes (Durack and Wijffels, 2010), is needed.

### 4. Summary and Conclusions

Analysis of annual maximum outcrop areas of the densest isopycnals to reach the surface in the northwestern North Pacific in winter indicates that there are significant interannual variations as shown by the time series of the $\sigma_\theta = 26.4–26.7 \, \text{kg m}^{-3}$ outcrop area in the EN4-OISST dataset starting in 1982 (Fig. 2). We consider the size of these outcrop areas as an indicator of how much ventilation is occurring at the bottom of the ventilated thermocline in the North Pacific since renewal of interior waters from the mixed layer and uptake of gasses only occurs when isopycnals outcrop. A significant 10-year lagged correlation exists between the $\sigma_\theta = 26.4–26.7 \, \text{kg m}^{-3}$ outcrop area variations and the ~bidecadal $O_2$ cycles at OSP in the northeastern North Pacific that exhibit the largest variability within this density range, giving support to the hypothesis that surface density variations are a main driver

of $O_2$ changes in the ocean interior. This is consistent with the finding by Sasano et al. (2018) that ventilation variability is a key process for generating $O_2$ trends and oscillations on isopycnals that outcrop in the North Pacific ($\sigma_\theta < 26.8$ kg m$^{-3}$; see their Fig. 11). The 10-yr lag corresponds approximately to the travel time of newly ventilated (and oxygenated) waters from the isopycnal outcrop locations in the northwestern North Pacific to OSP in the northeast where the isopycnals have descended to the subsurface. Kwon et al. (2016) found similar connections between isopycnal outcropping and ocean interior $O_2$ changes using the SODA model

whereas the results here are entirely data-based.

A new finding in our analysis is the dominance of SSS over SST in driving interannual fluctuations in annual maximum surface density (a ventilation proxy similar to annual maximum $\sigma_\theta = 26.4$–26.7 kg m$^{-3}$ outcrop area) in the northwestern North Pacific, as shown in Fig. 6b. Evidence that SSS variations in the northwestern North Pacific are occurring and play a major role in determining

interannual changes in upper ocean density and the related ventilation also comes from the analysis of SSS of shelf waters in the adjacent Sea of Okhotsk (Uehara et al., 2014). In contrast, SSS (freshening) and SST (warming) contribute about equally to the long-term declining density trends in the northwestern North Pacific (Fig. 6b) that are the cause of the long-term linear decline in $\sigma_\theta = 26.4$–26.7 kg m$^{-3}$ outcrop area and subsequently $O_2$ concentrations at OSP (Fig. 4) as well.

In contrast to Kwon et al. (2016), we do not find a good correlation between annual maximum surface density (or outcrop area) and thus ventilation and the PDO or other climate indices (NPGO, NPI). This may be because the analysis in this paper is using independent datasets and not a model, where forcing and ocean response are consistent by design. But given the dominance of SSS in determining interannual variations in surface density, a correlation with the PDO (which is based on SST) is not necessarily expected. While it is beyond the scope of this paper, we suggest for future work the development of a North Pacific climate index

that captures the SSS variability in the northwestern North Pacific as well as expansion of this work to include data-based subduction rates.

**Data availability**

The data used in this paper are available at the following web-sites:

EN4 - https://www.metoffice.gov.uk/hadobs/en4/

OISST - https://www.ncdc.noaa.gov/oisst

OISSS - https://podaac.jpl.nasa.gov/dataset/OISSS_L4_multimission_7day_v1

OSP - www.waterproperties.ca/linep/

CLIVAR/GO-SHIP sections - https://cchdo.ucsd.edu/products/goship-easyocean

PDO - https://www.ncei.noaa.gov/pub/data/cmb/ersst/v5/index/ersst.v5.pdo.dat

NPI - https://climatedataguide.ucar.edu/climate-data/north-pacific-np-index-trenberth-and-hurrell- monthly-and-winter

NPGO - http://www.o3d.org/npgo/

World Ocean Atlas climatology - https://www.ncei.noaa.gov/products/world-ocean-atlas

**Author contributions**

SM and KD designed the study and wrote the manuscript. SM was the project lead and made all the figures in the manuscript.

KD was the lead on analyzing the satellite data and combining them with other data products.

**Competing interests**

The authors declare that they have no conflict of interest.

**Acknowledgments**

This work was supported by NSF grant OCE-1851149. We would like to thank Roberta Hamme for input on a draft version of this paper, Tetjana Ross and two anonymous reviewers for their comments on the submitted manuscript, and Fisheries and Oceans Canada for making the OSP data publicly available.

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

**Figures**

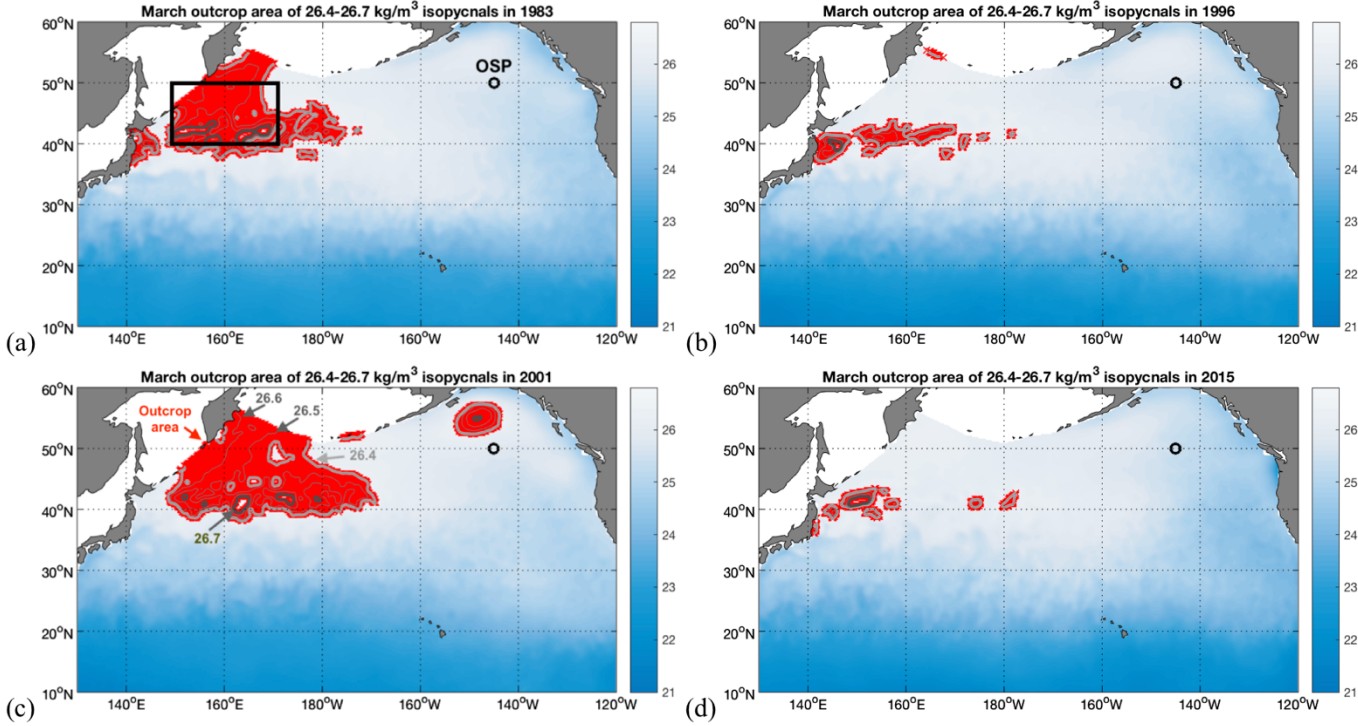


**Figure 1.** Examples of maxima (left) and minima (right) of March outcrop areas of the $\sigma_\theta = 26.4$–$26.7$ kg m$^{-3}$ isopycnal range: (a) 1983, (b) 1996, (c) 2001, and (d) 2015. Dataset used is EN4-OISST: 1/4° degree satellite SST combined with EN4 SSS (interpolated to 1/4°; see text). Color shading and colorbar are surface potential density anomaly, with anomalies in the $\sigma_\theta = 26.4$–$26.7$ kg m$^{-3}$ range indicated with red markers, highlighting the outcrop area of this isopycnal range as labeled in (c). Contour lines, as labeled in (c), show the $\sigma_\theta = 26.4$ kg m$^{-3}$ (light gray bold line), $\sigma_\theta = 26.5$ kg m$^{-3}$ and $\sigma_\theta = 26.6$ kg m$^{-3}$ (medium gray thin lines), and $\sigma_\theta = 26.7$ kg m$^{-3}$ (dark gray bold line) outcrops. The location of OSP at 145°W, 50°N, as labeled in (a), is marked by an open black circle. The area used for northwestern North Pacific SST, SSS, and surface density averaging, as shown in Fig. 6, A1, A8, A9, and A10, is outlined by a black box in (a).


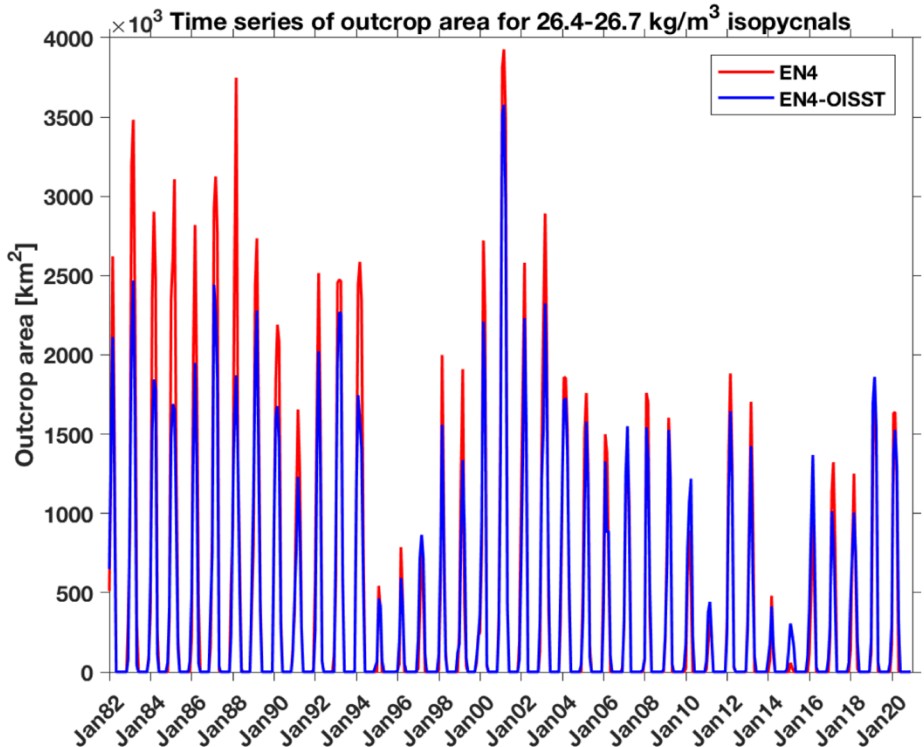

**Figure 2.** Time series of surface outcrop area of the $\sigma_\theta = 26.4$–26.7 kg m$^{-3}$ isopycnal range in the North Pacific from 1982–2020. Blue line is based on surface density from the 1/4° EN4-OISST dataset, and red line is based on surface density from the 1° EN4 dataset. The peaks in outcrop area occur just after January (shown on the x-axis), usually in March, of each year.

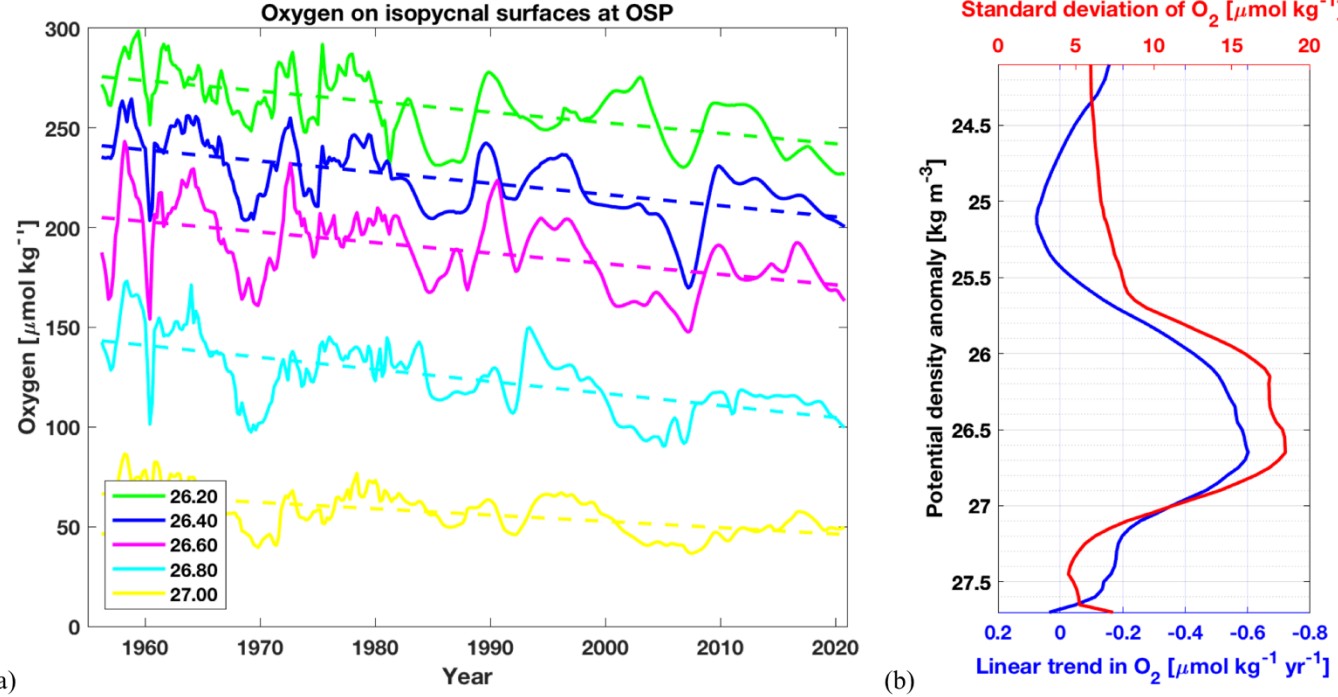

(a)                                                                                     (b)

**Figure 3.** (a) Time series of O$_2$ at OSP (see location in Fig. 1) on isopycnals from $\sigma_\theta = 26.2$–27.0 kg m$^{-3}$ (in 0.2 kg m$^{-3}$ increments). Linear trends of O$_2$ on each isopycnal are shown as dashed lines. (b) Isopycnal profile of the linear O$_2$ trends (blue) and the standard deviation of the O$_2$ variations (red) on isopycnals (in 0.05 kg m$^{-3}$ increments) at OSP. The vertical profiles were smoothed with a 9-point running average (corresponding to a 0.4 kg m$^{-3}$ averaging interval). Data used are objectively mapped O$_2$ data at OSP.

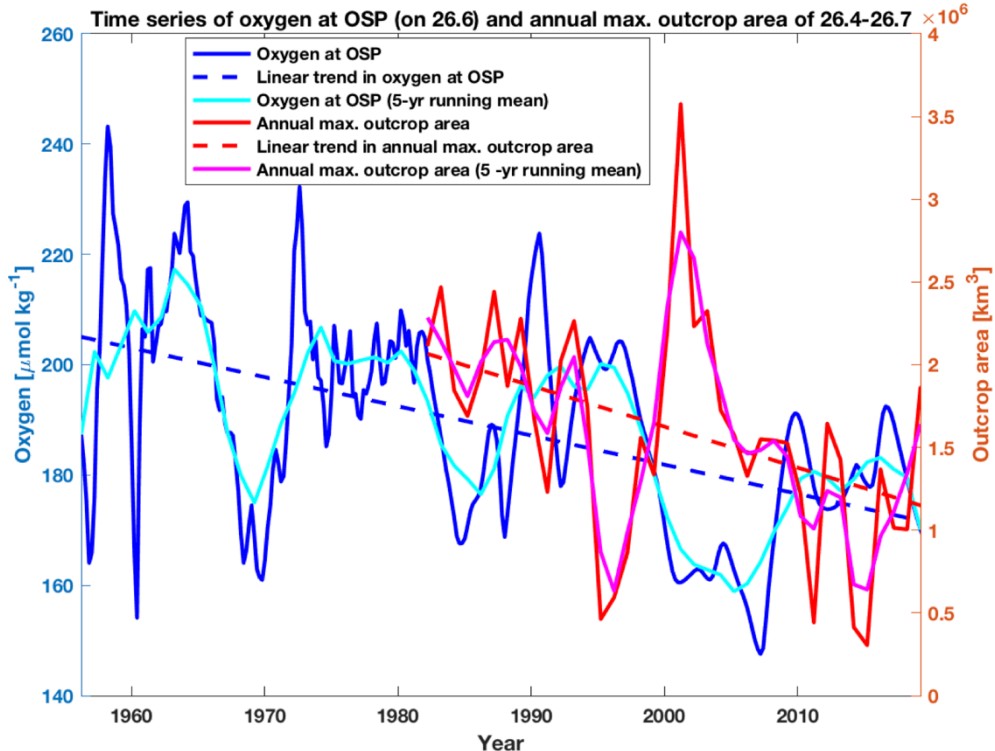

Figure 4. Time series of $O_2$ on $\sigma_\theta = 26.6$ kg m$^{-3}$ at OSP (magenta line in Fig. 3a) and of annual maximum outcrop area of $\sigma_\theta = 26.4–26.7$ kg m$^{-3}$ based on the EN4-OISST dataset (blue line in Fig. 2) plotted on top of each other (solid blue and red lines, respectively). Also shown are the data filtered with a 5-year running mean (solid cyan and magenta lines, respectively) and the linear trends in each time series (dashed blue and red lines, respectively). The linear trends are –0.53 µmol kg$^{-1}$ yr$^{-1}$ for the $O_2$ time series and –2.45 x 10$^{10}$ m$^2$ yr$^{-1}$ for the outcrop area time series.

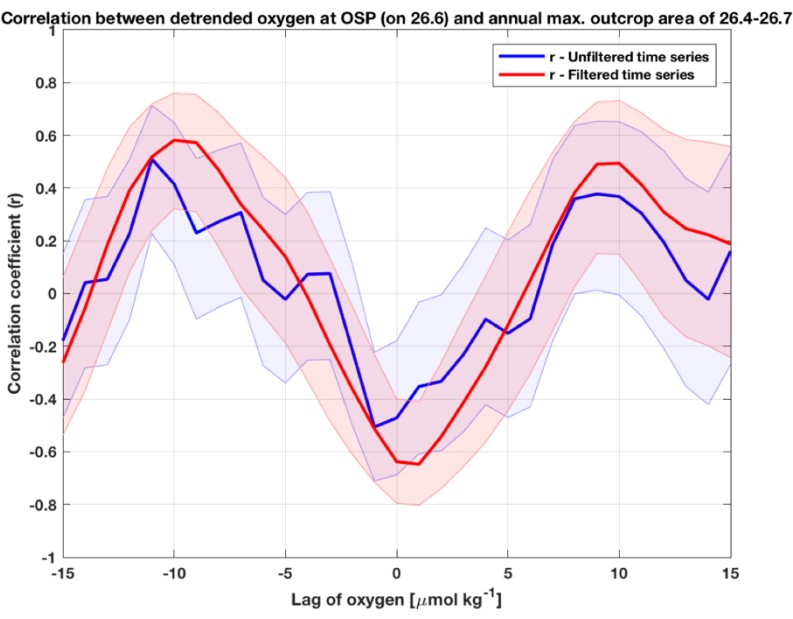

Figure 5. Lagged correlations between detrended time series of $O_2$ on $\sigma_\theta = 26.6$ kg m$^{-3}$ at OSP and of annual maximum outcrop area of $\sigma_\theta = 26.4–26.7$ kg m$^{-3}$ from EN4-OISST (solid lines in Fig. 4 with linear trends, the dashed lines in Fig. 4, removed), using unfiltered time series data (blue line) and the time series data filtered with a 5-yr running mean (red line). The x-axis represents the lag of the $O_2$ time series relative to the outcrop area time series. Shaded areas show 95% confidence intervals. Maximum correlation values (r) for the filtered data are 0.59 at a –10-yr lag and 0.50 at a +10-yr lag, maximum negative r-value (–0.65) occurs at close to a 0-yr lag.

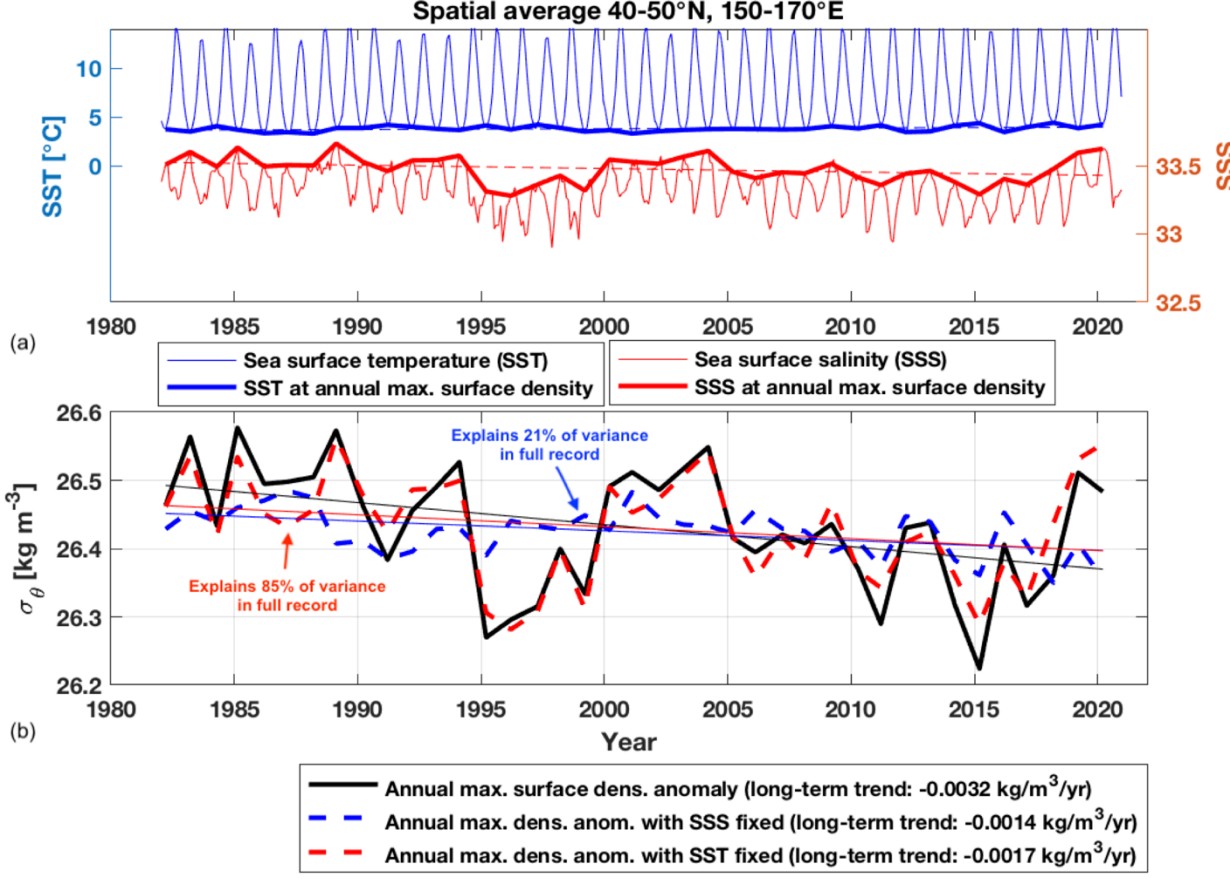

**Figure 6.** (a) SST (bold blue line) and SSS (bold red line) at annual maximum surface density, and (b) annual maximum surface density anomaly (bold black line) and contributions from SST (blue dashed line; with SSS fixed at mean annual cycle during density calculation) and SSS (red dashed line; with SST fixed at mean annual during density calculation) in the northwestern North Pacific (averaged over 40–50°N, 150–170°E; see Fig. 1a for area). The red dashed line (SST fixed) and blue dashed line (SSS fixed) in (b) explain 85% and 21% of variance of the full record (bold black line), respectively. Linear trends (dashed lines in (a) and thin solid lines in (b), using respective colors) are also shown. Thin solid lines in (a) represent monthly SST and SSS values. Dataset used is the EN4-OISST dataset.

550

555

**560**   **Appendix A**

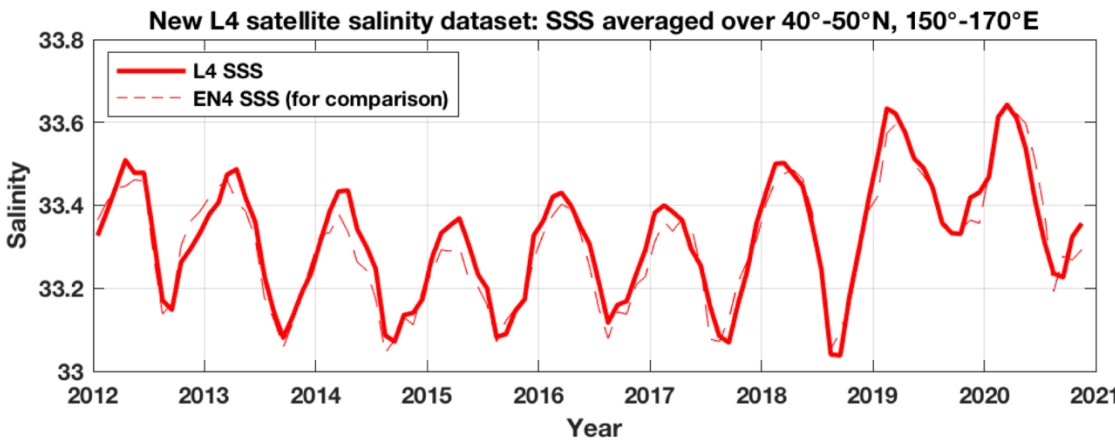

**565**   **Figure A1.** Satellite sea surface salinity (SSS) in the northwestern North Pacific averaged over 40–50°N, 150–170°E (see Fig. 1a for area) using the new Multi-Mission Optimally Interpolated Sea Surface Salinity (OISSS) Level 4 V1.0 dataset (Melnichenko et al., 2021). Also shown are SSS averages from the EN4 dataset (dashed line) for comparison.

**570**

**575**

**580**

**585**

**590**

**595**

**600**

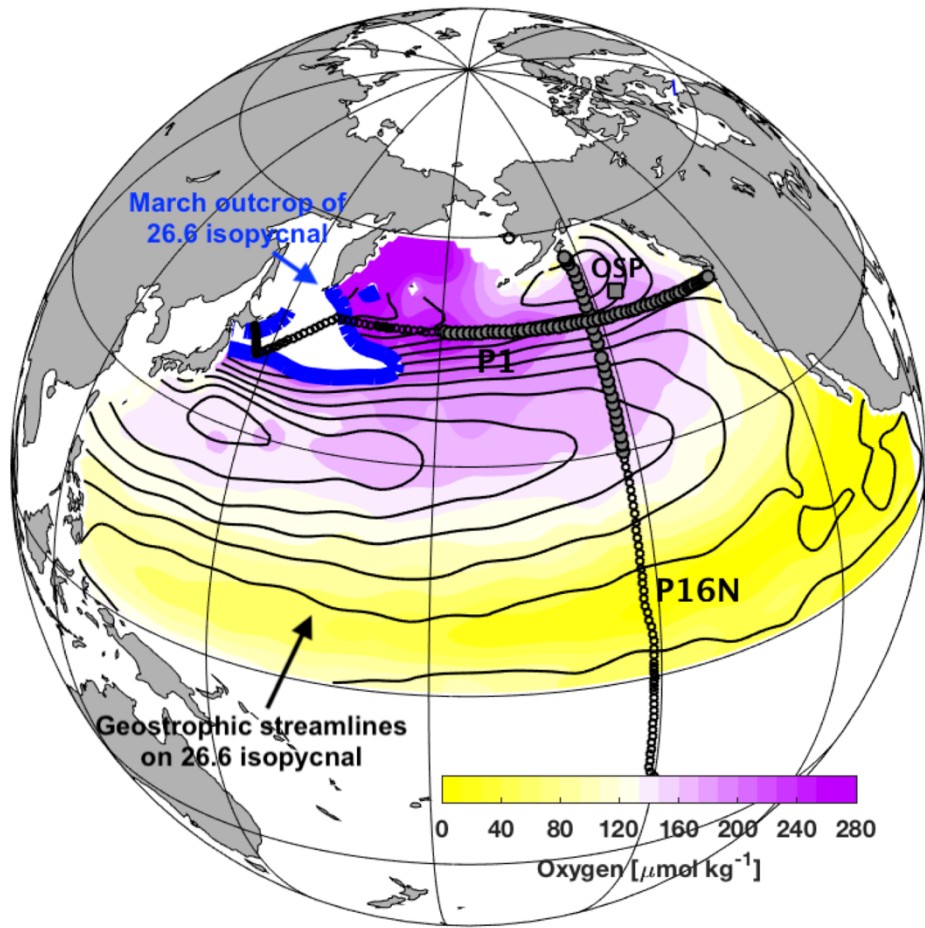

**Figure A2.** Dissolved oxygen (color shading) on $\sigma_\theta$ = 26.6 kg m$^{-3}$ in the North Pacific. Black contours mark streamlines (acceleration potential; Huang and Qiu, 1994) on the $\sigma_\theta$ = 26.6 kg m$^{-3}$ isopycnal, and bold blue contour marks the outcrop of this isopycnal in late winter (March). Oxygen decreases following streamlines because of respiration in the ocean interior. Data are from World Ocean Atlas and represent the climatological mean over six decades (Locarnini et al, 2013; Zweng et al., 2013; Garcia et al., 2014). In addition, circles mark station locations of the CLIVAR/GO-SHIP Repeat Hydrography cruises P16N and P1, with the larger gray-faced circles indicating the latitude and longitude ranges, respectively, of stations used for the difference plots in Fig. A6. The location of Ocean Station P (OSP; Whitney et al., 2007) is marked by a gray square.

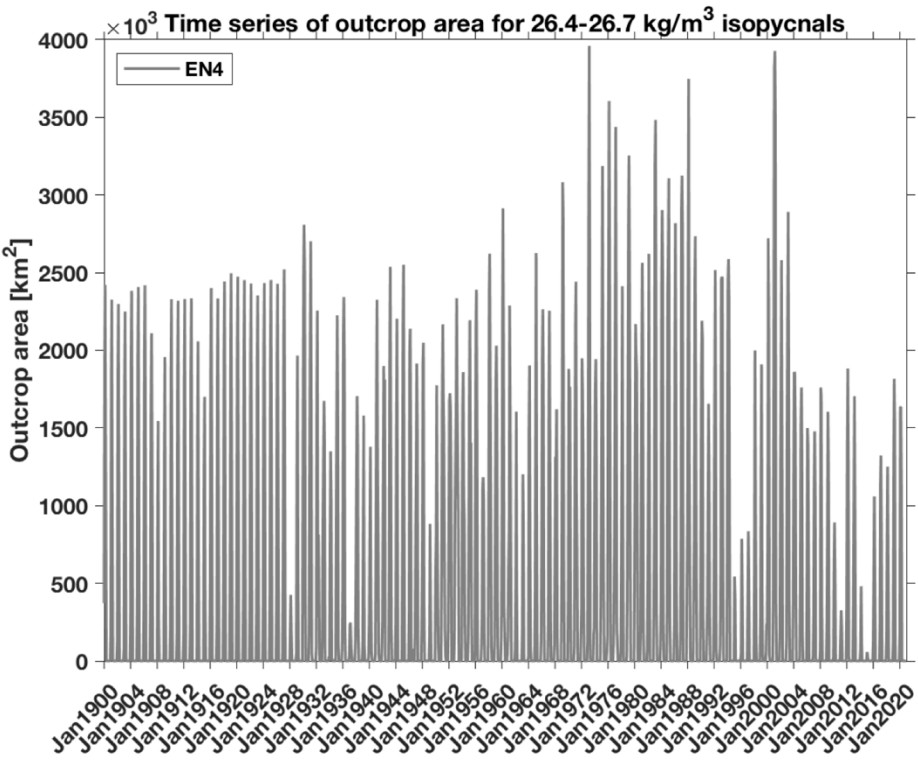


**Figure A3.** Time series of surface outcrop area of the $\sigma_\theta = 26.4$–$26.7$ kg m$^{-3}$ isopycnal range in the North Pacific, from the 1° EN4 dataset, for the full 1900–2020 record. The data since 1982 are the same as shown for EN4 in Fig. 2 (red line).







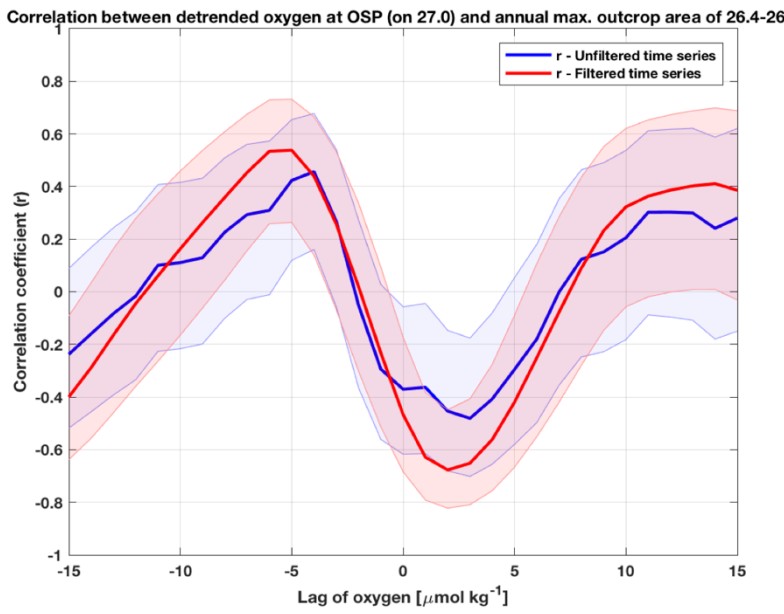

**Figure A4.** Lagged correlations between detrended time series of $O_2$ on $\sigma_\theta = 27.0$ kg m$^{-3}$ at OSP (solid yellow line in Fig. 3a) and of annual maximum outcrop area of $\sigma_\theta = 26.4$–$26.7$ kg m$^{-3}$ from EN4-OISST (red/magenta lines in Fig. 4), using unfiltered time series data (blue line) and time series data filtered with a 5-yr running mean (red line). Details follow Fig. 5.


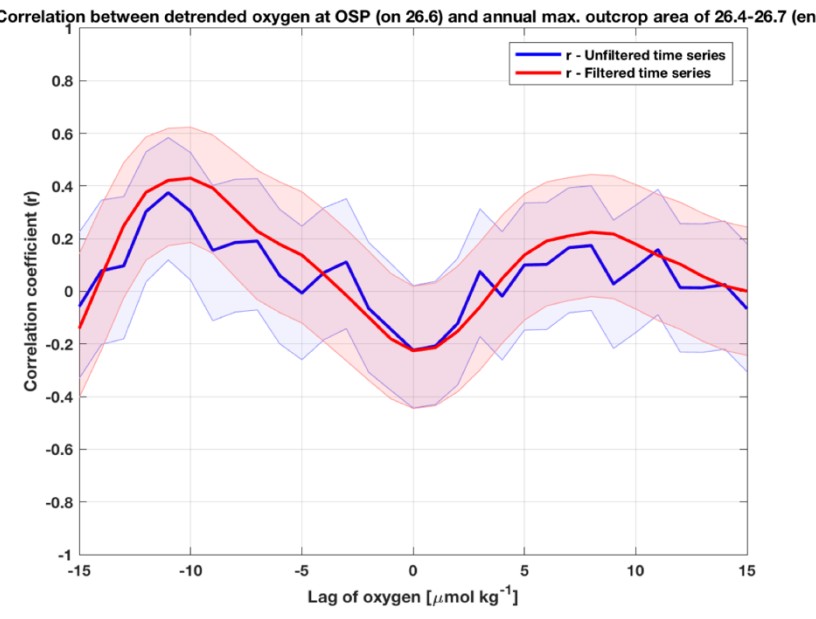

**Figure A5.** Lagged correlations between detrended time series of $O_2$ on $\sigma_\theta = 26.6$ kg m$^{-3}$ at OSP (1956–2020; blue/cyan lines in Fig. 4) and of annual maximum outcrop area of $\sigma_\theta = 26.4$–$26.7$ kg m$^{-3}$ from EN4 (using the 1941–2020 portion of the full outcrop area time series shown in Fig. A3), using unfiltered time series data (blue line) and time series data filtered with a 5-yr running mean (red line). Details follow Fig. 5.


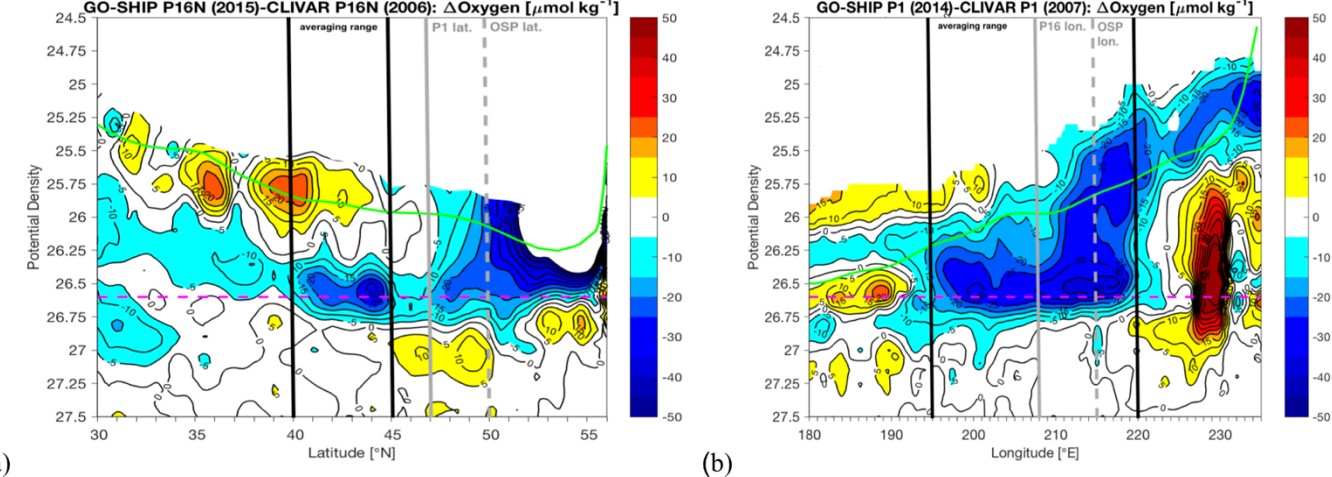

(a)     (b)

**Figure A6.** O$_2$ differences on potential density surfaces (color shading) for portions of two North Pacific repeat hydrography sections collected as part of CLIVAR/GO-SHIP: (a) along 152°W (P16N) – 2015 minus 2006, and (b) along 47°N (P1) – 2014 minus 2007. The locations of the sections relative to OSP are shown in Fig. A2. Changes are calculated by objectively mapping the O$_2$ data from each repeat occupation to create latitude/longitude versus density grids and differencing the mapped data. The dashed magenta line in each panel marks the $\sigma_\theta$ = 26.6 kg m$^{-3}$ isopycnal. The light green lines mark the March mixed layer density along the sections, calculated from climatological data. The latitude and longitude of OSP are marked in panels (a) and (b), respectively, as dashed gray lines, and the latitude and longitude where P16N and P1 cross are marked in panels (a) and (b), respectively, as solid gray lines. The two black solid lines in each panel mark the averaging intervals used for Fig. A7. Note that the averaging interval (40–45°N) in panel (a) is somewhat to the south of the OSP latitude because this marks the region of largest O$_2$ decline on $\sigma_\theta$ = 26.6 kg m$^{-3}$ in the interior away from the mixed layer and matches the averaging interval used in previous work (Mecking et al., 2008).

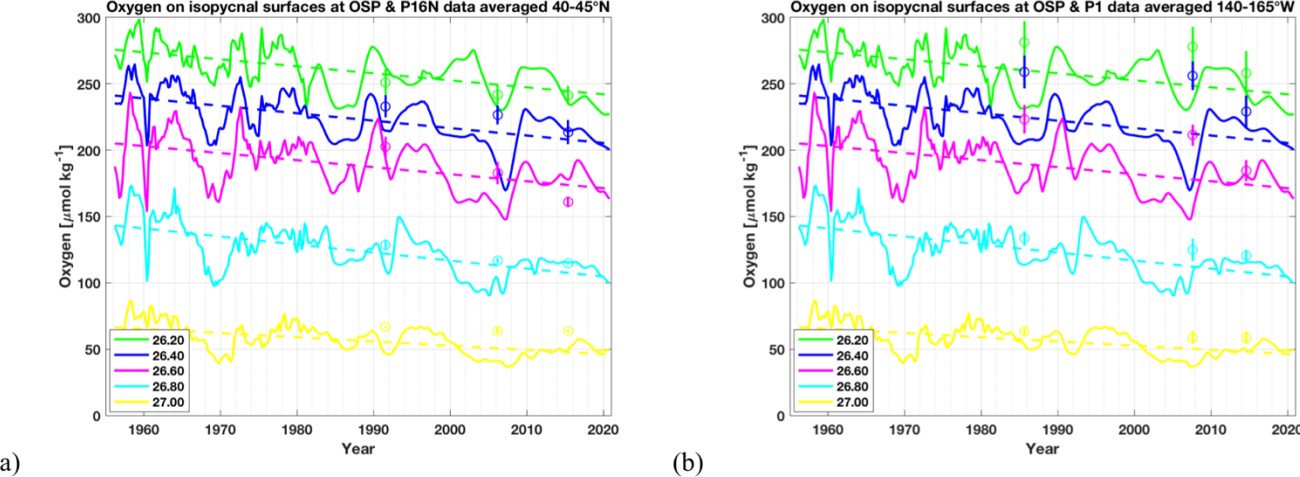

(a)     (b)

**Figure A7.** O$_2$ at OSP on isopycnals from $\sigma_\theta$=26.2-27.0 kg m$^{-3}$ (in 0.2 increments), including linear trends (dashed lines), as in Fig. 3a, together with averages on the same isopycnals from the WOCE (late 1980s/early 1990s), CLIVAR (2000s), and GO-SHIP (2010s) repeat hydrography cruises, indicated by circles (mean) and vertical lines (standard deviation): (a) using averages between 40–45°N for P16N cruises along 152°W in 1991, 2006, and 2015, and (b) using averages between 140–165°W for P1 cruises in 1987, 2007, and 2014. The latitudinal (P16N) and longitudinal (P1) averaging intervals are shown by the black lines in Fig. A6a and A6b, respectively.

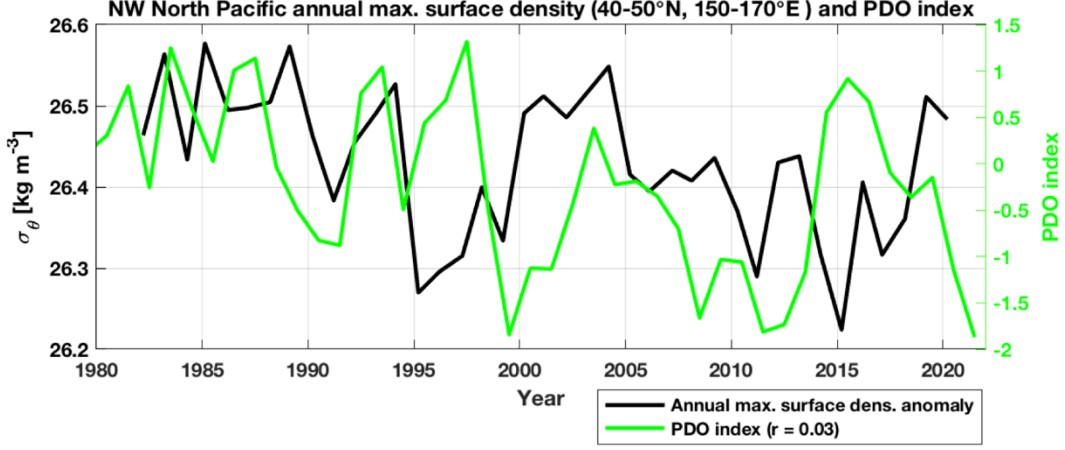

**Figure A8.** Annual maximum surface density anomaly in northwestern North Pacific averaged over 40–50°N, 150–170°E (black; same as black line in Fig. 6b) and annually averaged Pacific Decadal Oscillation (PDO; Mantua et al., 1997) index (light green).

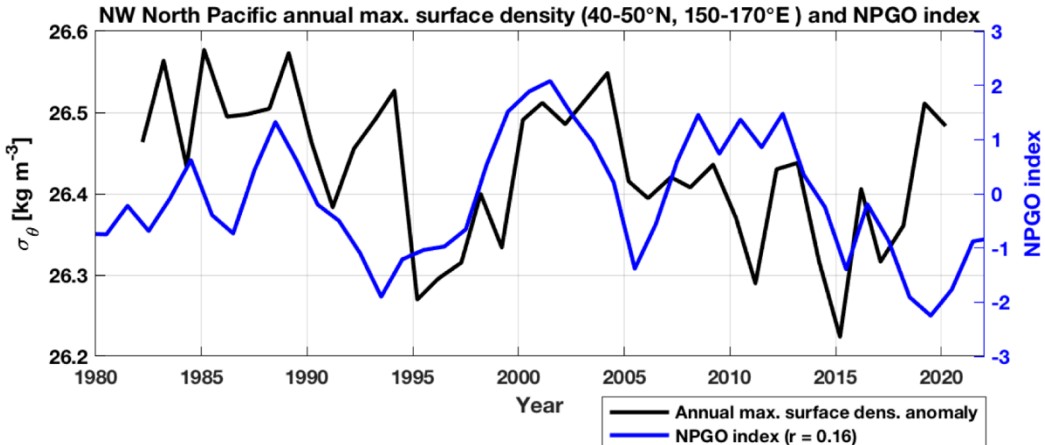

**Figure A9.** Annual maximum surface density anomaly in northwestern North Pacific averaged over 40–50°N, 150–170°E (black; same as black line in Fig. 6b) and annually averaged North Pacific Gyre Oscillation (NPGO; Di Lorenzo et al., 2008) index (blue).

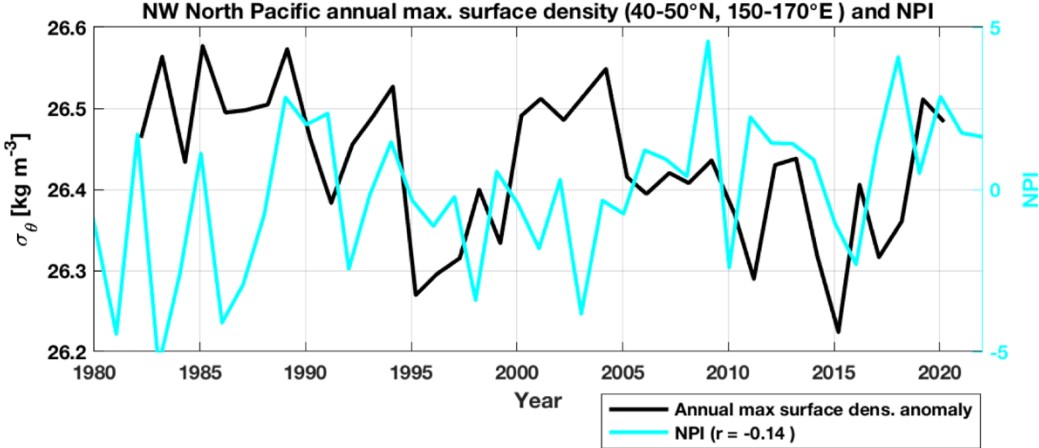

**Figure A10.** Annual maximum surface density anomaly in northwestern North Pacific averaged over 40–50°N, 150–170°E (black; same as black line in Fig. 6b) and annually averaged North Pacific Index (NPI; Trenberth and Hurrell, 1994) in cyan.