# Peer review of "Linking northeastern North Pacific oxygen changes to upstream surface outcrop variations"

_Biogeosciences, 2023_

## Author Comment (AC1)

**Author response to Reviewer #2 comments:**

Reviewer comments are in black, and our responses are in red.

I really enjoyed reading this manuscript. It is well-written, explores a simple idea very effectively and puts the results in context in a way that makes them potentially very useful.

I think this manuscript could be published as is (aside from a single typo) -- the figures are very readable and the text succinct and clear. However, I nevertheless will provide a few suggestions that I think would improve the impact of the manuscript.

Thank you for the encouraging comments. Below we provide responses to individual points raised and how we would incorporate suggestions when revising the manuscript (if invited to do so).

Line 39: Could also mention that similar cycles have been seen in DIC (e.g. Figure 7 in Franco et al 2021) -- this would allow you to pick up this point again at the end and suggest what you saw might also apply to CO2/DIC.

We appreciate the comment and will add a sentence that $O_2$ variability is closely linked to DIC (and nutrient) variability that has also been observed at OSP.

Line 235: I found the lagged correlations (Figure 5) really interesting as a way to assess how long it takes water to get from the NW Pacific to OSP (and the NE Pacific). My immediate question was: what does it look like for other isopycnals? It'd be powerful to see an increase with density and also really useful to have an idea of how fast signals are transmitted at different depth/isopycnals. I think you could make a nice comparison with Ueno & Yasuda 2003's Figure 7 (based on simple model).

Thank you for the suggestion. We looked at correlations between $O_2$ on isopycnals other than $\sigma_\theta$ = 26.6 kg m$^{-3}$ (lighter and heavier) and outcrop area - first using the same outcrop area ($\sigma_\theta$ = 26.4–26.7 kg m$^{-3}$) as used for the correlation in Figure 5 (line 235). For the lighter (i.e., outcropping) isopycnals we also investigated correlations with the outcropping area for densities bracketing the isopycnals themselves ($\sigma_\theta$ =26.3–26.5 kg m$^{-3}$ for $\sigma_\theta$ = 26.4 kg m$^{-3}$; $\sigma_\theta$ =26.2–26.4 for $\sigma_\theta$ = 26.3 kg m$^{-3}$). However, we do not find a depth trend in the $O_2$ lag for these lighter isopycnals, independent of which outcrop area we use. This is in part because the $O_2$ cycles on these isopycnals at OSP do not show any temporal offsets from one isopycnal to another (i.e. $O_2$ cycles on $\sigma_\theta$ = 26.2-26.6 kg m$^{-3}$ in Figure 3 are in sync with each other) even though travel times from outcrop area to OSP should be less on the lighter isopycnals. Our explanation for this is that the $O_2$ signal on the $\sigma_\theta$ = 26.6 kg m$^{-3}$ isopycnal is dominant because this isopycnal is at the bottom of the ventilated thermocline and the signal can be distributed to the lighter isopycnals through vertical mixing, thus the $O_2$ variability on the lighter isopycnals is in phase with$\sigma_\theta$ = 26.6 kg m$^{-3}$.

On the other hand, for the heavier isopycnals ($\sigma_\theta$ > 26.6 kg m$^{-3}$) we do find an indication that the lag between $O_2$ and $\sigma_\theta$ = 26.4–26.7 kg m$^{-3}$ outcrop area increases with depth. For example for $\sigma_\theta$

= 27.0 kg m$^{-3}$ , the best correlation for O$_2$ lagging outcrop area is at ~14 years as shown in the figure below (same as Figure 5 in the manuscript except that O$_2$ on $\sigma_\theta$ = 27.0 kg m$^{-3}$ is used):

[Figure]

This is qualitatively consistent with Figure 7 by Ueno and Yasuda (2003) which shows, as you mention, travel times increasing with density (from $\sigma_\theta$ = 26.7 kg m$^{-3}$ to $\sigma_\theta$ = 27.2 kg m$^{-3}$ in that case). We will mention this depth dependence of the lag where Figure 5 is discussed in the manuscript (lines 211–218) and can include the figure above in a revised manuscript (appendix) for illustration.

Line 342: Excellent point! It'd be really useful if you were able to suggest some ideas for what they might look like.

If we understand correctly, this comment refers to the suggestion on line 342 that "a climate index that better incorporates salinity is needed". We think that such an index should include E-P data from the northwestern North Pacific and/or surface salinity measurements at key entry points to the northwestern North Pacific. We will add some elaboration on the index at line 342 and in the conclusion section (where it also gets mentioned in the last sentence).

Line 347: Here's where I think the typo is. Is/are word/s missing? I find this phrasing confusing.

Yes, it should be "ventilated thermocline", not just "ventilated". Thank you for catching this typo. We will fix it.

**Citation**: https://doi.org/10.5194/bg-2023-132-RC2

---

## Author Comment (AC2)

**Author response to Reviewer #1 comments:**

Reviewer comments are in black, and our responses are in red.

This study attempts to understand the causes of decadal variability of dissolved oxygen in the Gulf of Alaska, observed at the Ocean Station P (OSP) at 145°W, 50°N. The oxygen timeseries at the OSP is the longest record of dissolved oxygen. The decadal variability of this data has been studied intensely over the last 20+ years and there are many hypotheses proposed to explain this variability. This study put forward the idea that the subduction of thermocline waters generates oxygen variability in the western Pacific, which would then propagate eastward following the circulation pathway of the North Pacific Current. Approximately 10 years later, the signal reaches the OSP and is observed there.

This hypothesis itself is not new, but what is new in this study is that a significant correlation was found between dissolved oxygen at OSP and the isopycnal outcrop area in the western Pacific which was reconstructed from the historical observations (EN4). Furthermore, the outcrop area is controlled by the density of the sea water at the surface, which further depends on the variability of sea surface salinity, rather than temperature. However, this variability does not exhibit significant correlation with the dominant mode of climate variability in this region. The maximum lag-correlation values are on the order of 0.5 to 0.6, which would explain about 25-36% of the total variance. This seems a significant contribution to the total oxygen variability, while it also leaves significant room for other mechanisms as well. My overall impression is that the manuscript contains significant progress on this problem and merits publication. Having said this, I would like to raise two points that the authors can consider for a revision.

Thank you for the comments. We appreciate the careful analysis of our manuscript. Below we provide responses to the two points raised and how we would address them when revising the manuscript (if invited to do so).

At L75, the authors choose to analyze data after 1982 only. I ask the authors to reconsider this choice because dissolved oxygen data from OSP exists from 1956. Since the focus of this study is the decadal variability with approximately 20-year timescale, the statistical significance of this analysis is critically limited by the effective sample size. The additional 26 years of data can capture additional full cycle of the signal potentially. Figure 2 indeed supports that the EN4-derived outcrop areas show the same pattern of maxima and minima as EN4-OISST. Then, it is possible to include the additional, extended timeseries before 1982.

Thanks for the comment. We did look at the full record of $\sigma_\theta$ = 26.4–26.7 kg m$^{-3}$ outcrop area calculated from the EN4 data since 1900 (in addition to the record since 1982 used in Figure 2) as shown here:

[Figure]

But we did find the relatively constant outcrop area in the early part of the record (1900-1925) questionable. While uncertainties are provided with each data point provided in EN4 due to gridding errors, uncertainties due to measurements error/bias do not appear to be fully accounted for (since uncertainties reported with the EN4 dataset do not become larger going back in time to 1900, as one would expect for earlier data, resulting in small uncertainties in outcrop area for the whole record). As a result, we concluded that the combination of OISST with EN4 SSS provided the most accurate and consistent dataset (at ¼ degree resolution) for us to use from 1982 (the start of OISST) onward. However, it is possible to also use more of the EN4 record as you point out (maybe starting 1925). We have recalculated the lagged correlations between outcrop area and $O_2$ at OSP as in Figure 5 in the manuscript, but using the EN4 record since 1941 for outcrop area (since the $O_2$ record starts in 1956 this is the first data year used when calculating correlations with a maximum lag of 15 years) instead of EN4-OISST. The result is shown here:

[Figure]

The patterns are the same as we had shown for the EN-OISST data in the manuscript (Figure 5) with maximum correlations at close to +/– 10 year lags, thus not altering our conclusions.

However, the magnitude of the maximum correlations is smaller than in Figure 5. For completeness, we suggest to add the two figures shown above (that use the EN4 data from before 1982) to the appendix of a revised version of the manuscript with elaboration and comparison to Figures 2 and 5 as they are now (using the EN4-OISST dataset since 1982) in the text.

At L135, it does make qualitative sense that a larger outcrop area indicates more ventilation, and a smaller outcrop area indicates less ventilation. However, not enough reasons were provided to justify why the spatial extent of the surface outcrop area is used as the only proxy for the amount of ventilation. There are atmospheric reanalysis data products available for estimating buoyancy and wind stress forcing. The latter can provide an estimate of Ekman flow. Mixed layer depths and geostrophic circulation can also be estimated from EN4 products. It makes me wonder if there were any reason that direct subduction estimates weren't used in this study. It is possible that these calculations were already performed by the authors or prior studies by others. If this is the case, it would be important to include this information.

We agree that calculation of subduction rates would provide a more complete metric of ventilation, i.e. of how much water is moved from the mixed layer into the thermocline within density classes. However, we find that using outcrop area as a proxy for ventilation is a simple first step since it only requires temperature and salinity data at the surface (compared to many different data products as you point to properly calculate subduction rates) and is thus easier to monitor, and since it is clear that there is no subduction if a density class stops outcropping. We will highlight these points more in the manuscript. We do point out at a couple of places in the manuscript (see lines 138–140, and lines 388–390 at the end) that proper calculation of subduction rates is a next step (which we have not done yet), but that it is beyond the scope of this paper. While vertical Ekman pumping and lateral induction estimates are needed to fully quantify the amount of water and $O_2$ that is transported across the base of the mixed layer (as carried out in a modeling study by Kwon et al. (2016) which suggested that outcrop area is the primary cause of interannual variability in subduction in the northwestern North Pacific), the spatial extent of the surface outcrop area can be used as an indicator for the amount of ventilation taking place. For clarity, we will make sure that a revised manuscript will emphasize earlier on that outcrop area is used as a simple proxy for ventilation (citing Kwon et al, 2016) and that more complex subduction rate calculations would be required to accurately describe ventilation.

**Citation**: https://doi.org/10.5194/bg-2023-132-RC1

---

## Author Comment (AC3)

**Author response to Reviewer #3 comments:**

Reviewer comments are in black, and our responses are in red.

**Review for "Linking northeastern North Pacific oxygen changes to upstream surface outcrop variations"**

**Overall**

The authors focused on investigating the temporal variablity of subsurface DO at the center of the eastern subarctic gyre (OSP), and the linkages to the surface density variabilities in the western subarctic gyres.
The OSP has been maintained for the longest periods in the global ocean, the partially updated discription was variable.
The mechanisms of the DO chnages shown in this work seemed to be raised in the previous studies, but the relationships between outcrop areas and subsurface DO changes were shown clearer in this study. In addition, the authors seemed to intend to describe carefully with comparison with previous studies and the other observations.
So I believe this work helps many researchers to udnerstand varialbilities in the northern North Pacific deeply, thus it's worth publishing after mminor changes.

Thank you for the thoughtful comments. Below we provide responses to the two points raised and how we would address them when revising the manuscript (if invited to do so).

Specific points.

1. The outcrop areas in the westeern subarctic gyre were broads especially around the periods of maxima. So all the points of the areas did not directly contribute to the DO variability. And it took some years to transfer the anomalies formed by the outcrop area changes even in the western subarctic gyres (e.g., Sasano et al., pointed out).
This may cause the trannsit time differences based on geostrophic currents and the lags outcrop areas and OSP DO chnages.
I think this is worth to add discussions.

Thank you. We will add references to Sasano et al. (2015, 2018) to the discussion of transit times, highlighting their findings that more than one process is affecting the $O_2$ cycles in the northwestern as well as northeastern North Pacific.

2. In the previous study (Kouketsu et al.; https://agupubs.onlinelibrary.wiley.com/doi/full/10.1029/2019JC015916), the figure similar to Figure A3b was shown. And they analyzes transfer the salinity anomalies from west to east, and showed the relationships nitrate changes. They discuss the changes transferring western subarctic gyres may transferring east of OSP, which seemed to be related to discrepancy between GO-SHIP and OSP changes pointed out in this study.
So I think it's worth citing in this study.

Thank you for the comment. We apologize for not including a reference to Kouketsu et al. (2020) who also show the 2014–2007 difference of $O_2$ along the P1 section among other properties (their Figure 4b). We will add a citation to that paper und include discussion of their analysis when addressing differences between GO-SHIP sections and OSP.

**Citation**: https://doi.org/10.5194/bg-2023-132-RC3